# Changing Tourism Trends of the Benedictine Abbey in Tihany: Best Practices of a Hungarian Monastic Community

**Dávid Fekete [1,*], Gábor Ágoston Barkó [2], Mihály Dömötör [3] and Katalin Czakó [4]**

1   Department for Regional Sciences and Rural Development, Albert Kázmér Faculty, Széchenyi István University, Egyetem Square 1, H-9026 Gyor, Hungary
2   Tihany Benedictine Abbey, I. András Square 1, H-8237 Tihany, Hungary
3   Doctoral School of Regional and Economic Sciences, Széchenyi István University, Egyetem Square 1, H-9026 Gyor, Hungary
4   Department of International and Applied Economics, Széchenyi István University, Egyetem Square 1, H-9026 Gyor, Hungary
*   Correspondence: fekete.david@sze.hu

**Abstract:** The touristic use of sacred sites is a widespread practice in Hungary. Throughout the past centuries, Christianity has dominated the history of the Carpathian Basin. The Hungarian State's strong affiliation with the Roman Catholic Church dates back to its foundation over a 1000 years ago. Due to its location on the shore of Lake Balaton and outstanding natural environment, the Benedictine Tihany Abbey is a major touristic destination and a popular place of pilgrimage in Hungary. The objective of the present paper is to examine how touristic activities contribute to the livelihood of a Hungarian monastic community and local economic development in the 21st century. To answer the research questions, the study primarily relied on literature and documentary analysis, in-depth interviews, and the construction and processing of a database. The research revealed that in addition to the classic elements (e.g., guided tours in the abbey, museum exhibitions, concerts, etc.), the program offer developed by the monastic community of Tihany is capable of addressing new target audiences: the rapidly rising number of tourists from the Far East (mainly Japan, China, South Korea, and Russia). Despite remaining considerably below visitor numbers recorded by the abbey in the mid-1990s, a significant increase in visitor numbers was observed in 2018–2019. The economic activities of the Benedictines of Tihany comprise a number of different areas beyond tourism, e.g., agricultural production and candle making, both of which are integrated into their tourism offer and feature among touristic programs. As a major employer, the abbey exerts a positive impact on the population retention capacity of the settlement by offering jobs to local residents, and it also contributes to population growth by attracting a large number of highly skilled professionals who choose to settle down in the region. The paper shows the evidence of the pandemic on pilgrimage and religious tourism in Tihany.

**Keywords:** religious tourism; Tihany Abbey; local economic development; Hungary; COVID-19



## 1. Introduction

The touristic use of sacred sites is a widespread practice in Hungary. Throughout the past centuries, Christianity has dominated the history of the Carpathian Basin. The Hungarian State's strong affiliation with the Roman Catholic Church dates back to its foundation over a 1000 years ago. Hungary is remarkably well-endowed in Christian churches and monuments, a heritage enriched by touristic values associated with various monastic orders. In recent years, the Hungarian tourism sector reached record high levels, including religious tourism. Despite the tourism sector being among the hardest hit by the global coronavirus epidemic, the situation of religious tourism deserves scientific scrutiny due to its significant role in the postpandemic recovery of affected areas.

Due to its location on the shore of Lake Balaton and outstanding natural environment, the Benedictine Tihany Abbey is a popular touristic destination and pilgrimage site in Hungary. It is an important hub for domestic and international tourism. When questioning a foreigner from a European country about places he has heard of or visited in Hungary, after Budapest he or she will mention Lake Balaton, most likely Tihany. The "downgrading" of Lake Balaton to the status of a touristic attraction and the Benedictine Abbey of Tihany to a museum would be overly reductionist, given the unique position of Lake Balaton among Hungarian regions in terms of its natural, geological, cultural, and architectural endowments. The role of territorial stakeholders, local governments, churches, and other organizations is to facilitate a more efficient utilization of socioeconomic resources and to implement the related development projects. The Benedictine Abbey of Tihany is organically connected to such heritage conservation and development activities.

Settlements such as Tihany located on the shores of Lake Balaton are uniquely positioned within the Hungarian settlement system from a spatial and economic development and an architectural and physical planning perspective alike. Their local economies are fueled by a high level of tourism and business tax revenues and a large proportion—the overwhelming majority—of their residents are employed in the tourism and hospitality sectors. This creates a specific "rhythm of life" for locals: a summer high season "overloaded" with working days and hours alternating with a winter season allowing plenty of time for rest and "cellar supplies". Tihany has also experienced the diffusion of this unique economic activity ("Balatonicum") and ensuing way of life since the 1960s. One of its consequences is the natural selection of the local population, manifest in the massive outmigration of those seeking work outside the tourism or hospitality sectors. Transforming this trend is a crucial prerequisite to the region's long-term viability and prosperity.

The migration of highly qualified people to large cities or foreign countries is a serious challenge. In domestic terms, Budapest and Győr (with its extensive network of public and church-owned schools, the Széchenyi István University, AUDI, county hospital, teachers, automotive engineers, doctors, etc.) exert the most significant draining effect. Efforts to reverse these unfavorable trends have been witnessed at the policy level since the 1990s, with the objective to retain the population of the more prestigious settlements on the shores of Lake Balaton and to encourage their long-term settlement. This large-scale regional development process consists of elements such as raising cultural standards, diversification of touristic programs, *providing year-round employment opportunities for the local population*, strengthening other economic sectors alongside tourism, and providing adequate employment and recreation opportunities for highly qualified individuals. It is underpinned by a special planning document prepared by the Balaton Development Council attached to the National Development and Regional Development Concept: the Development Program of the Balaton Priority Region, the thematic objectives of which are harmonized with the respective development concepts of settlements located on the shores of Lake Balaton (Völgyzugoly 2013).

The Benedictine Tihany Abbey significantly contributes to the above objectives through various local economic development tools and the hosting of nationwide cultural events. It is not a surprise: monasteries have been important places for economic activity for centuries, and nowadays monastic groups play an important role in local economic development processes (Isabelle 2023).

## 2. Objectives and Methods of the Research

The objective of the paper is to assess the contribution of tourism to the livelihood of a Hungarian monastic community and local economic development in the 21st century. For this purpose, three research questions have been formulated:

Q1—What characterizes the main features and adaptive capacity of the tourist offer of the Benedictine community in Tihany?

Q2—What major trends have been observed in the evolution of visitor numbers in recent years, with special regard to the COVID-19 era?

Q3—How does the Abbey's tourist offer affect the local economy and labor market?

The following methods were used to answer the research questions:

- literature and documentary analysis: assessing the role of Benedictine monks in regional development and the tourism activities they operate.
- in-depth interviews: to detect tourism trends of the past decades, five in-depth interviews were conducted with stakeholders dominantly shaping the tourist offer of the Benedictines in Tihany since the 1990s (three monks and two secular employees).
- database processing: the research undertook the summary of data on visitor numbers in the past few years. By entering guestbook entries into the database and their subsequent analysis, the paper was able to detect the number of non-paying visitors of the church (a separate database on paying visitors was available at the beginning of the research).

Since the present study is coauthored by a Benedictine monk in Tihany, special emphasis was laid on exploiting internal information, viewpoints, and impressions revealed by the in-depth interviews that had been neglected in previous scientific research. The involvement of a Benedictine coauthor implied a situational advantage whereby the identification and request of interviewees participation and access to necessary data and databases were rendered feasible (Appendix A).

## 3. The Contribution of Monks to the Local Economy and Tourism

The investigation of religious tourism can be observed in the field of social sciences for a long time. It is a touristic activity whose primary driving force is religious consideration, and this form of tourism is often realized during participation in a pilgrimage (Gisbert 1992; Rashid 2018; Solovieva 2022). It is one of the most ancient journeys of this kind, which has spread widely throughout the world over the centuries—regardless of the exect type of religion (Dallen and Olsen 2006). In many cases, it has reached such a scale in certain geographical locations that it requires special management, expertise, and marketing activities from religious organizations, for whom this is not their basic task (Razaq and Griffin 2015). Recently, several studies have addressed the impact of COVID-19 on religious tourism (Tsironis 2022; Roszak and Huzarek 2022; Hassan et al. 2022).

The monks' contribution to local economic development is a popular topic in international literature. Several studies have examined the contribution of monastic orders to the local economy in a historical approach (Li and Salonia 2020). Although our study deals with the Hungarian Benedictine monks, of course it can also be observed in the case of other religious orders that, during their local operations, whether through their tourism or specific economic activities, are significant and important players in the economic life of the region in that they run businesses and play a significant role in the local labor market (Roudometof and Michael 2010; Jonveaux 2019). According to some articles, monastic economy is able to show an alternative way to the market economy by emphasizing the importance of value-based economy and representing religious values in economic activities. Nowadays, monastic groups are good examples for these alternative ways of economic activity and producing (Isabelle 2011). In the literature, we rarely find examples of investigations related to monasteries in Hungary or Central and Eastern Europe (Niessen 2015; Anna and Michalkó 2013; Wiltshier and Clarke 2012; Isabelle 2021).

Due to their rich and long-standing history, the economic activities of Benedictine abbeys are highly diversified and structurally heterogeneous. Manual and intellectual labor have characterized the life of Benedictine communities since ancient times, which played a decisive role in laying the socioeconomic foundations of Christian Europe from the 6th–7th centuries onwards (Isabelle 2019). The governance of individual abbeys provided a pioneering model in organizational planning, dominantly shaping the evolution of Western European legal and economic systems (Emil et al. 2012). The success of Benedictine governance and management is underlined by their reputation as long-standing, economically viable, and stable institutions in several countries of various continents (Knesia et al. 2016). The monasteries have mostly specialized in agriculture and forestry, the organization of

local tourism, specific sectors of retail, sacral services, and various forms of real estate utilization (Silvia and Hiebl 2013). Benedictine abbeys are also well represented in the cultural sphere. They operate publishing houses, educational institutions, fine arts and library collections, and museums.[1]

In addition to the significant contribution of their economic activities to local and regional development, their cultural role enables Benedictine monasteries to "attract" and "address" wide segments of the population, positively impacting settlement-level community development, which underlines their complex role in shaping regional development. Whereas their economic development activities have a regional dimension, the impact of their community development and social organizational activities is limited to the settlement scale.

In her investigation of the structure and characteristics of monastic farming, Jonveaux (2019) found the fact that land and economic management have formed a part of the monastic life since ancient times to be somewhat at odds with their rejection of secular wealth, an essential manifestation of their way of life. The economies and new cultivation methods developed and practiced by monasteries have influenced the development of Europe throughout centuries, considerably raising the standard of European economic culture. Their farming activities allowed monasteries to establish ties and contact points with the external world, enabling them to reach out to wider segments of secular society. In addition, in the Western capitalist economic system, new cultivation methods introduced by monks provided an alternative, ecologically more sustainable and innovative way of farming.

The most relevant monastic activity from the perspective of the study is tourism, which cannot be treated separately from its local economic impact. Hence, by merging the two components, the study adopted a complex approach to community activities operated by the respective sacred site.

### 4. The History and Present of Benedictine Monks in Hungary

Hungary maintained strong ties with the Benedictine monastic order already in the period after the Conquest. Representing Hungarians who opted for Western Christianity, Prince Géza contacted Emperor Otto the Great in order to launch preparations for the conversion. A key figure of the mission was Bishop St. Adalbert under whose leadership the Benedictines from Brevnov moved to Hungary in 996. Géza and his son Stephen (the first Hungarian king canonized by the pope in 1083) designated the location of their monastery on the hill adjacent to the presumed birthplace of St. Martin, the future Pannonhalma. The martyrdom of St. Gellért in 1046 sealed the definitive settlement of the Benedictine Order in Hungary. This launched the beginning of an approx. 200-year period until the Tartar invasion, labeled justifiably in national ecclesiastic and cultural history as the Benedictine era. The 11th century witnessed the construction of royal abbeys governing Benedictine life in Hungary. In parallel, the proliferation of abbeys founded by various prestigious families and wealthy clans was already discernable under the reign of St. Stephen and particularly over the 12th century. They became the benefactors and protectors of abbeys established and maintained from the shared property of the clans (Csóka and Lajos 1970).

The Benedictine Order had already entered into decline in Western Europe by the 13th century, triggering as a response the launching of reforms and the establishment of new congregations later to be detached and organized into new orders. The Cistercian, Premontreian, and Augustinian orders emerged as the most influential reform orders whose spread and development fall outside the scope of the present study. The Benedictines had a paramount role in the diffusion of Christian culture in Europe and the Christianization of newcomer peoples. They laid the foundations of Western Christian scientific life and literature, invented Gregorian music and chant, and the Romanian architecture style. They supported rulers by advising them in matters related to the organization of newly formed states and by spreading more advanced forms and methods of farming among the peoples. The apparent exhaustion of the Benedictine Order due to its manifold and diverse activities and slowness to embrace new ideas and demands emerging in the 14th and 15th centuries

significantly retarded their development. Hence, leadership was taken over by other communities and cultural institutions (Söveges 1996).

While the principal cause of the decline of Benedictine presence in Western Europe in the 16th century was the spread of Protestantism, in Hungary it was triggered by the Turkish occupation. In the territories occupied by the Ottoman Empire, Benedictine abbeys were dissolved one after the other, while others were transformed into border fortresses, expulsing Benedictine life from within their walls. The majority of Hungarian territories had been liberated from the Turkish occupation by the end of the 17th century, and the removal of imminent danger and ameliorating financial conditions enabled the launching of much-needed construction works in the Benedictine Order and nationwide. Cofinanced from Austrian sources, the restoration of a number of ancient abbeys (Pannonhalma, Tihany, Bakonybél, Zalavár, Dömölk) had been concluded by the mid-18th century, triggering a revival of Benedictine life. In 1786, however, pursuant to a tolerance decree, Emperor Joseph II dissolved the Benedictine Order in the territory of his Empire in the spirit of Josephinism. In 1802, Emperor Francis I restored the Benedictine Order, assigning it a supplementary role in public education. This implied the takeover and own revenue-based operation of eight secondary schools with Benedictine teachers responsible for teaching and educational activities. Revenues from the holdings of the five abbeys (60,000 cat. acres) enabled the order to operate eight urban secondary schools—in Sopron, Győr, Pápa, Kőszeg, Bratislava, Trnava, Komárom, Esztergom, and later on Budapest and Pannonhalma in the place of Bratislava and Trnava—attended by 2000–2500 students annually. The nearly 150-years operation of this system was terminated in 1948–1950 when the Communist regime nationalized denominational schools and large estates and withdrew the operating license of religious orders. A convention concluded between the State and the Bishops' Conference in 1950 partially restored the operation of the shrinking Benedictine Order in the subsequent four decades of Socialism (Jánosi Gyula 1977).

Freedom of religion was reintroduced in Hungary post-regime change from the 1990s onwards, and the 1950 decree restricting the rights of churches was repealed by the new government. At the turn of the millennium, life was restored for the Hungarian Benedictines, this time integrated into European monasticism (Figure 1).

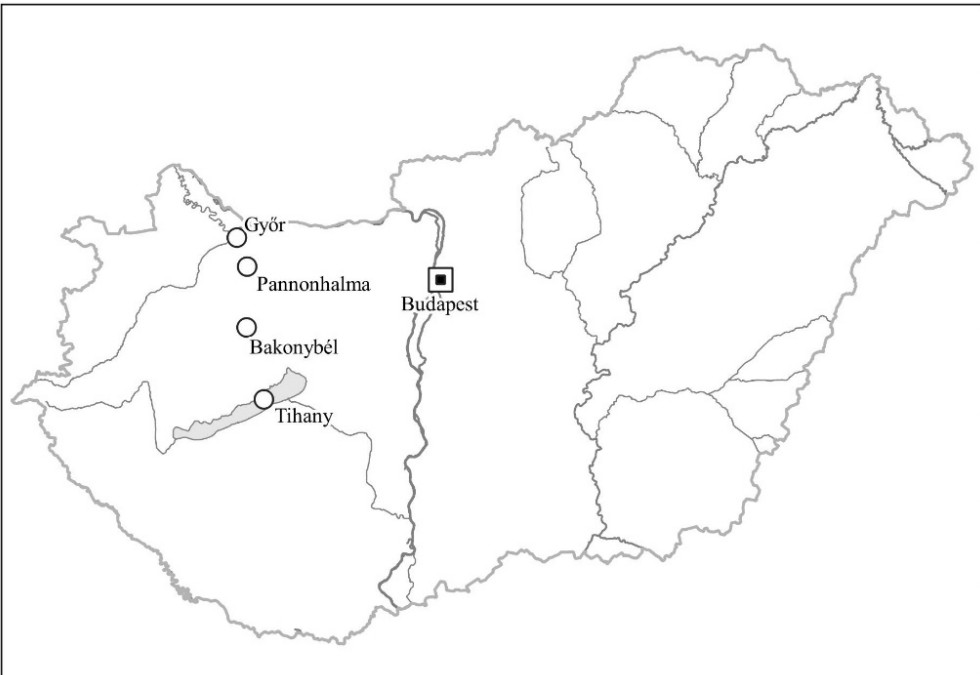

**Figure 1.** Benedictine Communities in Hungary. Source: Own compilatio.

## 5. Tihany Abbey

There are currently 11 Benedictine monks living in the Tihany Abbey (2019), a record high number in historical terms. The monastery was founded by King Andrew I (1046–1060) in 1055 as a family burial ground (monasterium proprium). The royal crypt beneath the shrine constructed in 1953 contains the skeleton of a man living in the 11th or 12th century (Uzsoki 1984). Monastic life was interrupted several times in the course of history: after the defeat at Mohács[2], the abbey was transformed into a border fortress (return in 1701), Hungarian King Joseph II dissolved the Hungarian monastic orders in 1786 (return from 1802), and the abbey was nationalized in 1950 under the Communist regime, eliminating the Benedictine presence from Tihany for over several decades (return: 1990) (Érszegi Géza-Barkó 2017).

A total of 80 secular workers are employed by the abbey and its affiliated institutions. The institutions belonging to the Tihany Abbey without independent legal status include:

- Saint Christopher Guesthouse,
- Hot kitchen for groups,
- Abbey Manor.
- The institutions with independent legal status include:
- Benedictine Abbey Museum,
- Tihany Roman Catholic Parish with two churches,
- Aszófő Roman Catholic Parish with five churches,
- Benedictine abbey Illyés Gyula Primary School and Elementary Arts Educational School,
- Abbey Rege Confectionery.

This staff number is significant in comparison to the population of a village of appr. 1350 people (Baunok et al. 2011), given that the abbey offers its employees *year-round employment*, a non-negligible factor in the Lake Balaton area. As an ecclesiastical site, an imperative of conscience obliges them—as well as a special emphasis on the role of families and large families in their teachings—to organize the economic management of their property with an eye on ensuring year-round employment for the maximum number of secular persons.

Prior to the nationalization of 1950, the abbey owned land-holdings of about 6900 hectares, whose centralized management in Pannonhalma left property rights intact, since inalienability was guaranteed by their acquisition as a royal donation. In the past, property donated by Hungarian kings would fulfill the function of special-purpose assets: the revenues of abbeys of the Order of Saint Benedict of Pannonhalma (today: Hungarian Benedictine Congregation) ensured the maintenance, among others, of the abovementioned 8 secondary schools with 3271 students enrolled in 1945 and people's schools with 6503 students in 1945 (Csóka 2020). Currently, the abbey is the owner of one 7-hectare land-holding on the northern shore of the Inner Lake in Tihany or the southern slope (which is barely a thousandth of its pre-nationalization land property!), facilitating the cultivation of 40,000 stems of lavender and 15,000 different herbs. In a few years, the land will play a significant role in the diversification of their economic activities and revenues.

## 6. Changing Tourism Trends and Frameworks

The outstanding tourist environment designates tourism as the abbey's major economic activity and source of income. Nonetheless, the sector has undergone dramatic restructuring in the last thirty years. According to the unanimous view of the subjects of in-depth interviews, during the previous "classic" or "retro" periods[3] visitors to Lake Balaton considered bathing a priority and their stay extended to several weeks, while nowadays the average duration of stay has been reduced to one day or the period of a long weekend, and (paying) visitors' preferences have increasingly shifted to cultural programs comprising higher intellectual value added (e.g., Gregorian chant performed by skilled singers, preparing herbal tea blends led by a specialist), drawing on ancient traditions of the site (silence, peace, tranquility, monks) or other unique and extraordinary programs.

Up until the end of the 1990s, the Abbey Church and Museum in Tihany were visited almost "automatically" by tourists relaxing at Lake Balaton. For instance, in the year 1996, the 1100th Anniversary of the Hungarian Conquest, the Abbey Church and Museum had 396,000 visitors. This number refers to those guests who did not come to participate in spiritual programs (mass, prayers) but purchased entry tickets, but the number of "guest believers" attending liturgical services was also significant. Nowadays, it is no longer natural for a Hungarian primary school student or interested tourist vacationing at Lake Balaton to visit the (supposedly) "unchanged" abbey in Tihany, which is much more than an ancient church. Data on visitor numbers from recent years clearly demonstrate that the number of visitors in the mid-1990s considerably exceeded their current number, despite a sharp rise observed during the two years preceding the outbreak of the coronavirus crisis (Figure 2).

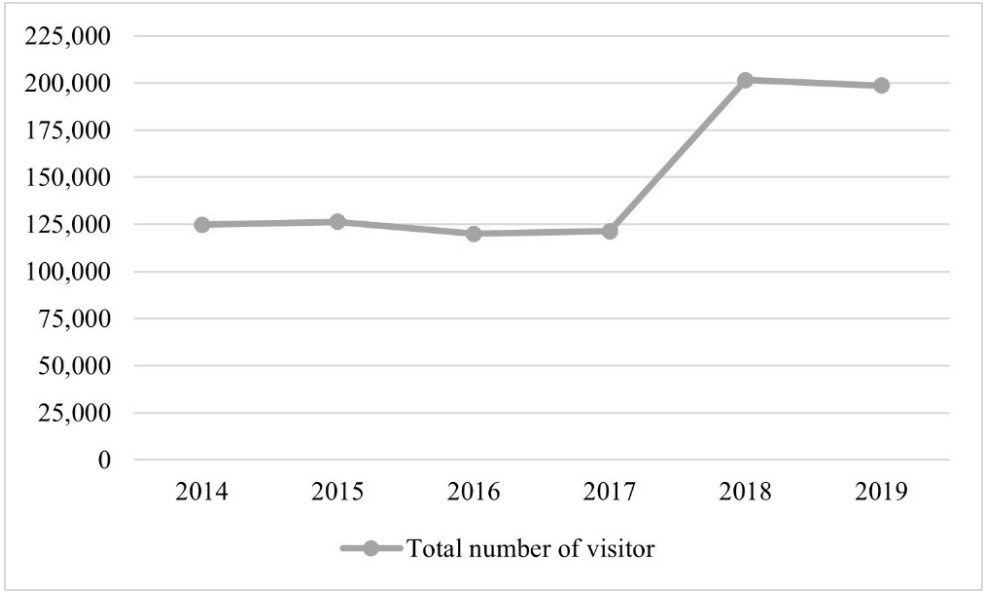

**Figure 2.** Evolution of the number of visitors of the Benedictine Abbey Church and Museum in Tihany between 2014 and 2019. Source: Own compilation.

The results of interviews also indicated changes in the distribution of visitors by place of origin. In addition to Western (Austrian, German, French, Italian) tourists, a growing number of tourist group arrivals from Russia, Ukraine, and the Far East, mostly from China and Korea, have been registered. Finding appropriate channels to address the latter essentially relies on showcasing culturally neutral signals that may offer visitors a memorable experience (e.g., the sunrise/sunset orientation of the Tihany Abbey and medieval holy places, the importance of the Sun in the East and the West, the silence of the peninsula, the symbiosis of the natural environment and the built architecture). Albeit to a lesser extent than in the Far East and the "Russian world", the abbey has become increasingly popular among tourists from Visegrad countries over the recent years. Addressing the differential needs of various target groups (e.g., "beach tourists", pilgrims, Russians and Ukrainians seeking to reinforce their national identity) has become a crucial priority. Programs and exhibitions offered by the abbey are available all year round, which has a considerable impact on the local economy, as guests visiting the settlement outside the summer high season provide extra revenues for local businesses (e.g., shops, accommodation).

The abbey's program offer was revised in full consideration of the above processes facilitating increased tourist turnover and income. The annual number of visitors purchasing entry tickets is approx. 130–150 thousand, which is significant in domestic terms (in the Balaton region, only the Keszthely Helikon Castle and the Castle of Szigliget outnumber the abbey in terms of visitors).

While the preserved crypt (Subchurch) of King Andrew I dating back to the 11th century and the Museum and Gallery of Modern Arts hosting different exhibitions yearly can be accessed through the Abbey Church, the visitor does not have to purchase a ticket to the house of God, the temple. Anyone signaling their intention to pray at the gate is admitted by the staff and may profit from a moment of prayer and recollection while sitting on a bench. Hence, in addition to touristic valorization, the uninterrupted provision of the original ecclesiastic function is also a priority. The operation of the museum receives no state funding, and the Order has to cover regular maintenance, renovation, operational, and wage-related expenses from the museum's own revenues. A common misconception among people is that a church museum benefits from the funding provided by the Hungarian state to church schools and social institutions, i.e., a "head quota" quasi fully covering the operational costs of a given institution, but this is not the case for museum institutions.

## 7. Effects of COVID-19 on Tourism and Visiting Numbers

COVID-19 pandemic exploded as well as in Hungary in March 2020 and it created a difficult situation for the actors of tourism at both domestic and international level. In the beginning of the promising season, strict closings and central dispositions that limit every section of life were plugged by the governments in Europe. The tourism, as a section, builds mostly on the possibilities and advantages of modernization, and it was forced to stop due to the epidemic measures. The touristic actors had no other choice but to survive with the using of reserve sources. There was no difference regarding sacral touristic and pilgrimage destinations; the decrease caused by the pandemic was forcibly seen on the visitor numbers, which caused financial problems in the annual operating budget.

The Benedictine Abbey in Tihany is a determinative and symbolic touristic site of the Balaton, visited by hundreds of tourists and pilgrims each year. The main attraction of the abbey is the Abbey Church and Museum, and a big decrease in visitor numbers was seen as well on the annual numbers. Beside the numerous domestic tourists, there is a determinative number of visitors from European countries, such as Germany, France and Italy, and during the summer period a significant number of them visits the Benedictine Abbey in Tihany. It is very important that the number of the visitors has grown from Russia and the Far East Region in the last few years. Visitors from China, South Korea, and the so-called ASEAN countries (Vietnam, Laos, Indonesia) are dominant principally in the last group. We see from this tendency as well that the possibilities of the abbey were completely changed by the strict closings.

It is possible to call 2019 the last year of peace when the visitor number of the Abbey Church and Museum was above 150,000, which proved to be a stable annual visitor number in the last two decades. In the following two years, in 2020 and in 2021, the tendency was changed in consequence of closings and restrictions with the result of reduced visitor number. The abbey was visited in 2020 by 74,238 visitors, in 2021 by 77,184 visitors.

> *According to the border closures and traveling restrictions, there were no Russian, Ukrainian, and Southeast Asian groups, and the foreign guided tours were missing in general. For two years, there were no school trips, so the number of the school groups visiting the abbey was minimized.* (Benedictine monk of Tihany)

The above-mentioned school and children's groups used to be dominant in the period from May to June and in September, with the effect of expanding the summer high season. Considering the domestic tourism in 2020, there was an explosion in July that lasted until mid-September (because the former strict restrictions were moderated significantly after the first wave of the pandemic), but not in 2021; therefore, there was a huge decline in the number of the domestic tourists during the two years (Figure 3).

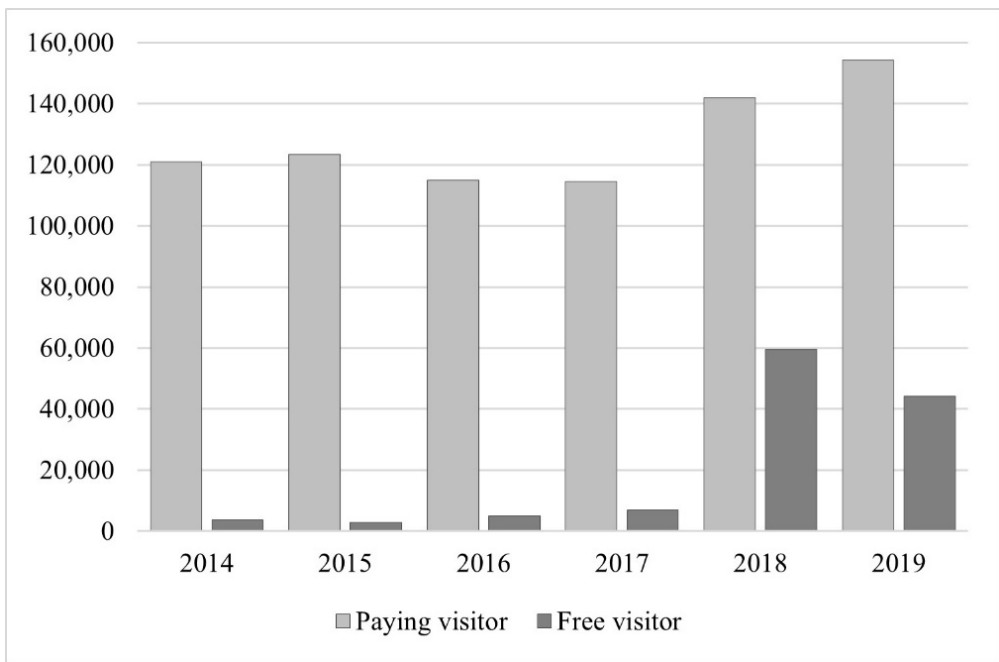

**Figure 3.** Evolution of the number of visitors of the Benedictine Abbey Church and Museum in Tihany between 2014 and 2019, according to visitor type (paying or free). Source: Own compilation.

The annual absolute visitor numbers also show the decline, which characterized the abbey's attendance over the two years. The "hopeful" start in 2020 was followed by a big decline, and then a relatively high number of guests visited the abbey in summer, which fell to virtually zero as a result of the end-of-year epidemic measures (Figure 4). The restrictive measures in 2021 lasted until the end of April, so the opening in May could only uneasily moderate the zero values of the first months, and the summer season was not very strong.

> *This is also due to the fact that, compared to 2020, Hungarian people were braver to travel abroad, so they did not prefer the destinations of Balaton as they did in the previous summer. Compared to the previous year, the modest attendance figures of the last months of 2021 also gave cause for hope.* (secular employee of Tihany)

The rate of decline in the number of visitors is perhaps more nuanced than the absolute number of it, both in nominal and percentage terms. We examined the decline of the two years by first comparing the decline to the base year 2019, and then comparing the two years affected by the epidemic. The largest decrease in terms of quantities was in the number of visitors between the summers of 2019 and 2021, followed by the difference between the summer explosion of 2020 and the previous summer season. It is encouraging that the last four months of the relation 2021–2020 already show a positive divergence, which is expected to continue in 2022 (Figure 5).

For 2022, there were high hopes for free travel, which could bring a boom in external group travels and class trips. Unfortunately, domestic and foreign policy developments, such as the escalating Russian-Ukrainian conflict, will also have a major impact on tourism scenarios (Benedictine monk of Tihany).

Another positive outcome of the epidemic period for the Abbey Museum was the availability of time and skilled staff to create new interactive exhibitions. This allowed the renewal and renovation work to be achieved without visitors.

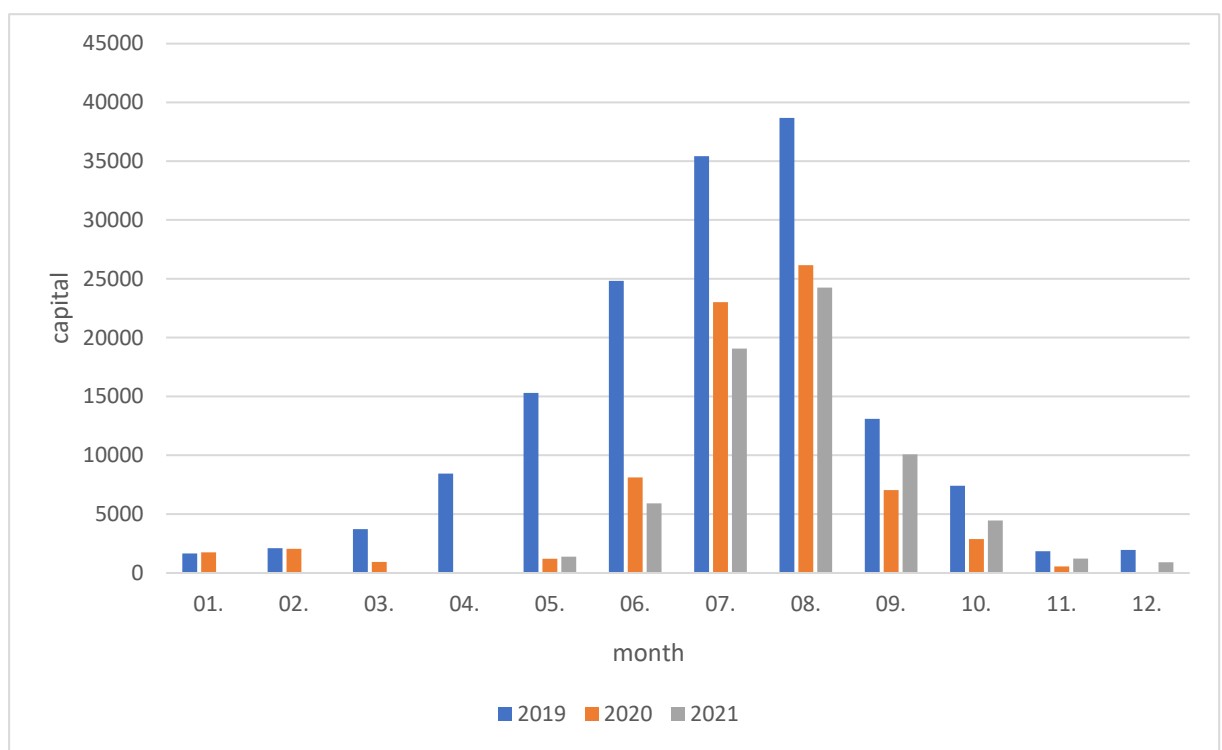

**Figure 4.** Visitor numbers in 2019, 2020 and in 2021. Resource: own design based on the dataset of the Benedictine Abbey in Tihany.

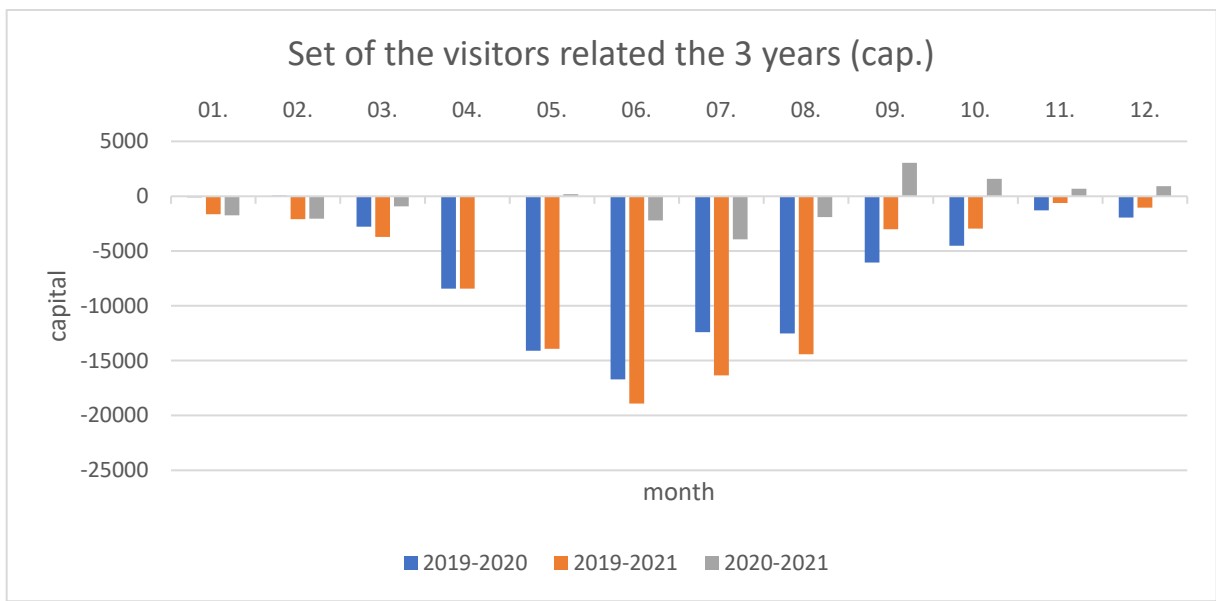

**Figure 5.** Decline in the number of visitors. Resource: own design based on the dataset of the Benedictine Abbey in Tihany.

## 8. A New Approach to Complex Touristic Offer

The aim of this section is to present novel best practices developed lately by the monastic community of Tihany in the field of tourism. In addition to addressing new, potential target groups beyond the abbey's traditional visitors, these measures may significantly contribute to local economic development in a small settlement with a heavy reliance on tourism.

**9. Museum Programs, Tours**

The traditional offer of the Benedictine Abbey Museum allows visitors access to a trilingual information board in the baroque church, entry to the royal crypt, and historical and fine art exhibitions. These can be visited individually or in groups, under the guidance of a monk, which enjoys widespread popularity. Thematic programs complement this basic offer since experience suggests that monastic life and the monastic community attract considerably more attention than classic elements such as historical museum exhibitions. Such programs include the aforementioned monk-led tours, the abbey Adventure Tour, the program entitled "Sounds of Silence" as well as "The Lights of Dawn". During the approx. two-hour long nocturnal tour called *"Sounds of Silence"*, the silence of dark halls and interior spaces normally inaccessible for tourists can be explored under the expert guidance of a Benedictine monk. Tickets for this unique program usually sell out months in advance. *"The Lights of Dawn"* is a unique walking tour and musical experience allowing visitors to await sunrise from 04.50 a.m. in the summer months, which attract a large number of visitors from all over the country. Visitors joining the early morning walking tour guided by a Benedictine priest are offered a glimpse into the most sequestered spaces of monks, the everyday lives of their 1000-year-old community. At the program's start, participants may enjoy the first morning sunbeams from the fathers' spice garden overlooking Lake Balaton, and when the lights reflected on the surface of Lake Balaton turn into gold and Tihany begins to rise from its sleep, they continue their journey toward the incense-laden Subchurch. While the first rays of the rising sun illuminate the tomb of Andrew I, the group listens to the ethereal Gregorian chant of Benedictine students, followed by a walk to the baroque church, where in the morning silence they can discover the path that leads to their interior before the hustle and bustle of the day begins. At the closing of the program, monks open the Abbey Rege Confectionery where guests may enjoy the breathtaking panorama of Lake Balaton and the delicacies of monastic gastronomy, including abbey-made beverages (Tihanyi 2019).

Museum developments from project funding were completed during the winter period of 2019–2020 (Balaton 2020). Since the spring of 2020, an interactive exhibition showcases the history and current practice of local monastic life focusing on the basic rudiments of Christian faith presented in a comprehensible and modern language, i.e., subjects "springing up" from monastic life have been displayed within the abbey's walls. The exhibition is titled *"Soul over the Waters—the Meeting of Past and Present in the Tihany Abbey"*. The illustrative presentation of the Deed of Foundation of Tihany Abbey, the architecture of the medieval monastery, experiencing the natural and spiritual milieu of Tihany in the exhibition space, or performing monastic activities (preparation of herbal tea blends, candle making), are all related to this thematic subject. In addition, a Sunrise Terrace in the garden of Abbot Lazarus offering a panorama over the eastern basin of Lake Balaton promotes the realization of the objectives of the "Lights of Dawn" walking tour. The aim is to allow visitors to gain a better insight into local Benedictine monastic life and to experience the outstanding natural environment of Tihany, as these are the major attractions. Not surprisingly, in the 19th century, the Tihany Abbey was referred to as "the crown of Lake Balaton".

The realization of new museum attractions introduced in 2016 required extra staff, experts, or students enrolled in general training institutions to support the manager of the museum, the Benedictine monk with a degree in humanities. Hence, each summer (from May to August), 4–6 students from the Benedictine Grammar School in Győr, preferably graduates with plans to pursue their higher educational studies in the domain of music, language, humanities, law, or natural sciences, are employed. These young people tend to return each year, enriching programs with their expert knowledge and participating in the education of the next generation. The direction of the Tihany-Győr migration trend, at least for the duration of the summer season, has thus been reversed in the Benedictine context by recruiting young people from rapidly developing cities or urban agglomerations for

tasks that demand cognitive, creative, foreign language, and communication skills. It is not uncommon for these young people to prolong their stay in Tihany for over a year.

Experience demonstrates that a very low proportion of young people with a "humanities background" or "humanities literacy" enrolled in secondary or higher educational institutions in the area of tourism and hospitality have adequate linguistic sophistication and negotiation skills and marketing talent and, in addition to conversational-level English, basic knowledge of a European language, and last but not least, advanced social and communication skills.

## 10. Abbey Rege Confectionery

Among the touristic sights of the abbey, the Rege Confectionery deserves special attention, distinguished among other famous confectionaries in the country by its exceptional panorama ("Royal Lodge of Hungary"). Opened in 1961 and undergoing extensive modernization in 1996–1997, it has been refurbished over the last three years: in 2016, its operation was transferred to the abbey, and it welcomes its guests with a hot kitchen from the spring of 2017, along with an interior with a panoramic window from the spring of 2018, and a completely upgraded selection of locally prepared confectionaries, cold and hot dishes from the spring of 2019. The extension and renovation of the Saint Christopher Guesthouse located next to the Rege Confectionery will be undertaken in the coming years. For the time being, it is suitable primarily for families and groups of pilgrims and students, capable of accommodating 40 people in 9 rooms.

In addition to the approximately 15 employers of the Rege Confectionery, the heart of the place, the kitchen employs a nationally renown master confectioner with permanent residence in the area. The first chef (2016–2019) arrived from Budapest from a popular restaurant with a regular appearance on gastro-blogs, and he settled in the vicinity of Tihany and envisions to spend the rest of his life there.

## 11. Concerts

Each summer, the Abbey Church's Rege courtyard situated in front of the Rege Confectionery and boasting excellent acoustics provides the venue for a nationally renowned concert series, which greatly enhances the cultural life of the area and attracts classical music lovers to Tihany. Following the Austrian pattern, some concerts are combined with a champagne toast in the inner garden of the abbey, creating a unique atmosphere. Since 2019, Tihany has also been home to "Klassz a pARTon", the festival of national significance attracting visitors from all over the country.

According to the interviews, the positive trends in the development of musical life allowed for the recruitment of two professional, internationally renowned organists/church musicians currently teaching in the music school of the Benedictine School in Tihany. They are also planning to establish their residence in the town or the nearby area.

## 12. Abbey Manor

In addition to tourism, the abbey pursues various other closely related economic activities. These are mainly located on the above-mentioned 7 hectares of land, situated in an exceptional natural environment along the western fringe of the village on the shores of the Inner Lake. The land was returned to the abbey in 1994. Its special value lies in the fact that it belongs to the original land donation dating back to the reign of King Andrew I. For over 23 years, it served as a pasture for horses. In 2017, 40,000 lavender and 15,000 herb plants were planted (Figure 6). Their primary utilization is the commercial distribution of manufactured products. Herbs are sold by the abbey in the form of various tea blends, while lavender can be purchased in the form of plants and oils. This constitutes an important economic development activity due to its reliance on labor-intensive extensive horticulture. After drying, plants are being processed all year round. Future plans include the construction of a manor for the location of processing plants (dryers, oil presses), and a showroom where interested visitors can taste the products made from various plants.

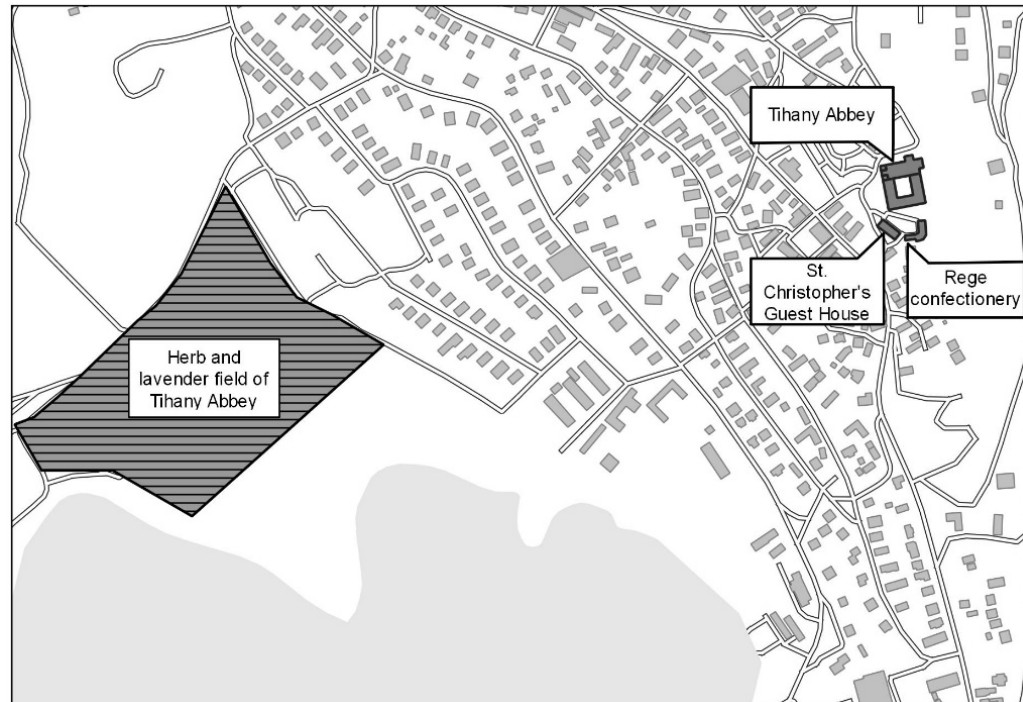

**Figure 6.** The touristic attractions and economic units of the Benedictine monks of Tihany. Source: Own compilation.

The St. Florian's Candle Making Workshop, currently operated in the tiny cellar of the abbey, will also be housed in the complex designed in full harmony with the landscape. The current underdevelopment of the craft of candle making necessitates important investments. This provides temporary employment opportunities for 2–3 people. In recent years, lavender and related products have attracted a great deal of attention, thanks to the 7-hectare land with its visitor center serving as an important market and demonstration venue. Since the spring of 2018, a qualified horticultural engineer from the Great Plain has been employed year round, who has already founded a family in Tihany.

**13. Pastoral Service, Teaching**

Two further activities are of outstanding importance for the abbey and the monastic community: pastoral care and teaching. These are of course basic activities of the monastery, so not a new approach. It should be still mentioned, because the Benedictines have an effect on the region through these activities as well. The Tihany Abbey is connected to two parishes, Tihany and Aszófő. The latter has incorporated four extra villages (Balatonudvari, Fövenyes, Örvényes, Balatonszőlős). The organization of local community life is an important task and opportunity for these settlements. A settlement where collaboration exists among local leaders and the appointed pastor are conducive to vibrant, healthy, and cheerful communities that local young people are eager to join, strengthening their desire for rootedness. It is important to emphasize that two among the six affected settlements—Balatonudvari and Örvényes—are potential locations for young couples benefiting from the Hungarian government's generous child and large family support program (Magyarország 2019). The Benedictine Abbey of Tihany is involved in several cooperation programs with the Municipality of Balatonudvari.

Teaching, a fundamental activity related to the Benedictine way of life, dominates the everyday life of the Tihany community. The abbey operates the elementary and music school. The number of students exceeds 180. Outside Tihany, children from 17 settlements, i.e., a 40 km radius area (!) attend these schools. In the life of a settlement, a vibrant school is a strategic asset due to its contribution to shaping the future of children and forming the next generations of the settlement's residents. For this reason, the school offers

a wide spectrum of special activities for students (e.g., musical education, folk dance, sports opportunities, study groups, summer camps), enriching the cultural life of the settlement and encouraging students' participation in higher education in areas that also correspond to local job opportunities (e.g., teacher, tourism expert, musician, agricultural engineer).

## 14. Discussion

In the course of history, Benedictine monks been influential players in a great variety of domains. The situation is no different today, as demonstrated by their role in the operation of the major tourist destination of Hungary's second most visited tourist area. The survey has identified a number of best practices that enrich the repository of efficient solutions for the operation of similar tourist destinations. It is important to highlight that the Tihany Abbey places great emphasis on offering visitors new types of programs in addition to the classic tourist offer. Best practices include, e.g., the "Voices of Silence" event that allows participants to abandon the hustle and bustle of everyday life, and the chief attraction for 21st century visitors with a hectic lifestyle is found to be the tradition, activity, and way of life practiced by monastic communities, such as the Benedictines for centuries. Through such programs, as stated by one of our monastic interviewees *the Benedictines of Tihany were able to identify and address new target groups such as* e.g., *the metropolitan society of managers, most of whom had not been particularly receptive to sacred messages and sites.*" (secular employee of Tihany)

Another noteworthy target group is the community of tourists from the East, represented in steeply growing numbers in Hungary and Tihany, particularly preceding the outbreak of the coronavirus. The Benedictines of Tihany adjusted their program offer to their needs by focusing on the symbols connecting Eastern and Western religions. *"The use of this system of symbols indicates that emphasizing common denominators is an efficient means to foster dialogue between different world religions.*" (Benedictine monk of Tihany) However, this is not the only area where, in addition to earthly opportunities, the sacred site represents otherworldly obligations for the Benedictines of Tihany: " . . . *we never collect entrance fees from those who enter the church to pray. The house of God is open to all*" (Benedictine monk of Tihany).

Another important finding of the paper is that more recent economic activities such as agricultural production, candle making, etc. are closely related to tourism. Hence, in the case of a major tourist destination, the fact that a church community carries out economic activities does not imply an absence of linkages with their touristic profile. Accordingly, lavender fields, which benefit the Tihany Abbey by the sale of produce, also represent a tourism dividend as a basis of their marketing strategy, and the lavender harvest has grown into an important touristic event. It is important to underline the role of the Benedictines of Tihany as a major employer in the area, a significant asset for a settlement with an overwhelming reliance on seasonal workers. However, as highlighted on several occasions in the study, the monks also employ individuals who are experts in their field but are not locals, which enables them to contribute to population growth in the area of Tihany by increasing the rate of locally available, qualified workforce.

Finally, it has to be underlined that the COVID-19 pandemic caused a huge decline in the visitor numbers of the abbey. In 2019, the Abbey Church and Museum were visited by 150,000 people, while in 2020 and in 2021, the tendency changed in consequence of closings and restrictions with the result of reduced visitor number. In the COVID-19 era, the abbey was visited only by approx. 75,000 visitors a year, so reached only the half of the visitor number of 2019. Nowadays, the Russian-Ukrainian war is causing another problem because tourists from these countries will be missing for long time. The abbey used the COVID-19 period for creating new interactive exhibitions in the museum and this period secured a "calm" time for the renewal and renovation works without visitors.

### 15. Conclusions

The study reviewed the tourism activities of the Benedictine Abbey in Tihany. The apparent abundance of sacred hotspots in Hungary is explained by the role of the Hungarian state as an important base for Christian denominations during the 1000 years since its foundation. The history of the Benedictine Order in the Carpathian Basin dates back to over a millennium, and their role and presence has been continuously expanding in recent centuries. The history of the Benedictine Abbey in Tihany, albeit with several interruptions, also looks back to a thousand years, and in the contemporary era, it is an important center of the Benedictine Order in Hungary and an outstanding tourist destination. This latter role stems from the position of Lake Balaton as the second most visited tourist hotspot in Hungary after Budapest, and the abbey' strategic location in the region. In the beginning of the research, three research questions were raised (What characterizes the main features and adaptive capacity of the tourist offer of the Benedictine community in Tihany? What major trends have been observed in the evolution of visitor numbers in the recent years, with special regard to the COVID-19 era? How does the abbey's tourist offer affect the local economy and labor market?) The methodology of the study primarily relied on literature and documentary analysis, the construction and processing of a database, while in-depth interviews revealed useful information that had not been published in previous studies.

According to the literature review, it is worth analyzing monastic orders and their local embeddedness: those communities used to be important actors of local economy and society and they are still playing important role regarding these areas (Marcin 2020).

An important objective of the paper was to identify the key features of the tourism offer of the Benedictine Abbey of Tihany as a sacred tourist destination and to assess the institution's capacity to respond to new challenges in tourism and to address new target groups. The research demonstrates that, in addition to the classic elements (e.g., guided tours in the abbey, museum exhibitions, concerts, etc.), the program offer of the monastic community of Tihany is capable of addressing new target audiences, i.e., the steadily increasing number of tourists from the Far East (mainly Japan, China, South Korea, and Russia).

The research also revealed visitor numbers of recent years by tourist type, distinguishing between paying and non-paying visitors. Despite remaining considerably below visitor numbers recorded by the abbey in the mid-1990s, a significant increase in visitor numbers was observed in 2018–2019 in parallel to the introduction of the revised program offer. While the COVID-19 pandemic has practically eliminated foreign tourists from the Tihany Abbey, they are expected to return in the postpandemic period.

In summary, after the regime change, there were several significant trend reversals in the tourism life of the Tihany Abbey (based on interviews with monks and touristic experts):

Period 1: "Socialism that lives on" 1994–1999. In 1994, the Benedictines became the caretakers of the abbey and the museum again. During this period, the "legacy of socialism" was still alive, and many Hungarian and Central and Eastern European tourists did not have the money to go abroad, so they chose the most popular domestic destination, Balaton and with it Tihany. In the new Benedictine era, the largest number of visitors can be observed in 1996, the year of the Hungarian millennium: 396,000 people visited the abbey.

Period 2: "Hungarian tourists are heading abroad" 2000–2005. From the end of the 1990s, the financial possibilities of Hungarians increased, so the number of visitors decreased continuously because more and more people began to travel abroad, and due to the decrease in the population, fewer and fewer school classes visited the abbey. This is an important factor because Tihany is included as an almost mandatory class excursion route in Hungarian public education (general and secondary education).

Period 3: "The period of the economic crisis" 2006–2012. The next decline began in the mid-2000s, when the financial possibilities of the population decreased from 2006 onwards, and in 2008 an economic crisis broke out. In such cases, the population in Eastern Europe always saves on culture, first on vacation.

Period 4: "Increasing number of domestic and international visitors" 2012–2020. A trend reversal followed in the mid-2010s. The number of visitors began to increase modestly but steadily: the Hungarian government's policy of "opening up to the east" was also felt in Balaton tourism: Russian, Ukrainian, and Southeast Asian groups (ASEAN countries) began to arrive in significant numbers, and from the mid-2010s significant economic growth took place every year, so the number of visitors from the domestic population also increased. Until the mid-2000s, the abbey's foreign guests in terms of his/her country of origin, Germany was in the leading position (memory of GDR/Germany's meetings), but now it is not decisive, or only pensioners come. However, this is typical for the whole of Western Europe: this kind of "conservative v. church" attractions are mostly chosen by the generation with a somewhat religious background and upbringing, who are now definitely over 60 years old. In terms of Hungary, the abbey is still popular among students, but since there are fewer and fewer school classes due to the shrinking population, there are fewer and fewer student tickets, etc.

Period 5: "Period of continuous crises" 2020 to the present: (COVID, Russian-Ukrainian war, energy crisis, etc.) During the pandemic, Russian, Ukrainian, and Southeast Asian tourists almost disappeared; not a single Russian or Ukrainian group has arrived since the start of the war. There are still strict COVID restrictions in Southeast Asia, especially China, so essentially no groups have arrived from there in 2022. In 2021 and 2022, Hungarian tourists absolutely dominated. Based on the interests, they are looking for exclusive programs (tours enhanced with gastronomy, special tours at night and at dawn, tours with musical experiences, etc.); companies, groups, and individuals are willing to come even in a crisis, just make it special! This trend is expected in 2023 as well.

The research results demonstrate that the economic activities of the Benedictines of Tihany comprise a number of different areas beyond tourism, e.g., agricultural production and candle making, both of which are integrated into their tourism offer and feature among touristic programs. As a major employer, the abbey exerts a positive impact on the population retention capacity of the settlement by offering jobs to local residents; in the meantime, it contributes to population growth by attracting skilled professionals willing to settle down in the region.

In this study, we present what innovative program elements the Tihany Benedictine community has added to its touristic offer. In doing so, this study also used new approaches: in addition to mass tourism, the community is increasingly open to the more qualified groups requiring high-quality programs (e.g., managers, company leaders, etc.), who, in addition to the usual tourist attractions, also enjoy a spiritual experience at the abbey's innovative programs (e.g., Sounds of Silence, Lights of Dawn). An important new approach is that the offered programs are not classical church ceremonies but have a spiritual task and content and they convey Christian values, the possibility of spiritual replenishment, and spiritual deepening to non-believers as well.

**Author Contributions:** Conceptualization, D.F.; methodology, D.F. and K. C.; software, M.D.; validation, D.F., M.D. and G.Á.B.; formal analysis, D.F. and G.Á.B.; investigation, K.C.; resources, G.Á.B.; data curation, M.D.; writing—original draft preparation, D.F., M.D. and G.Á.B.; writing—review and editing, D.F.; visualization, M.D. and K.C.; supervision, K.C.; project administration, M.D.; funding acquisition, D.F. All authors have read and agreed to the published version of the manuscript.

**Funding:** The APC was funded by Széchenyi István Egyetem Publikáció Támogatási Program.

**Informed Consent Statement:** Informed consent was obtained from all subjects involved in the study.

**Data Availability Statement:** Data available on request due to restrictions privacy.

**Conflicts of Interest:** The authors declare no conflict of interest.

## Appendix A. List of Interviewees

1. J.M., 35, prior of the community, monk (script, 2021. 07. 07., Tihany, Hungary)
2. Á. B., 37, monk, touristic director (script, 2021. 09. 22., Tihany, Hungary)

3.	Á. M., 29, monk, responsible for international issues (script, 2021. 09. 22., Tihany, Hungary)
4.	B. K., 44, secular employee, visitor center (script, 2021. 08. 18., Tihany, Hungary)
5.	M. V., 52, secular employee, manager, confectionery (script, 2021. 08. 18., Tihany, Hungary)

## Notes

1	Keplinger et al., Entrepreneurial activities.
2	In 1526, the Ottoman Empire won a major battle against the Kingdom of Hungary at the settlement of Mohács, marking the beginning of the 150-year-long occupation of the vast majority of Hungarian territories by Ottoman troops.
3	The beginnings of mass "beach tourism" can be traced back to 1960, the opening of the Kis Tihany Hotel in Tihany.

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
