# Peer review of "Changing Tourism Trends of the Benedictine Abbey in Tihany: Best Practices of a Hungarian Monastic Community"

_religions, doi:10.3390/rel14040435_

Round 1

Reviewer 1 Report

First of all, I would like to commend the author for taking up an interesting topic - certainly the place described in the article is worth citing as an example of sacred cultural heritage and its adaptation to tourism.

The premise of the submission, its objectives and methodology were carefully explained and correctly formulated. I have no objections to this part in principle. 

My comments are basically directed to three elements:
1. The theoretical underpinnings of the study are fairly poorly presented. Especially in the chapter "The contribution of monks to the local economy and tourism" it would have been worthwhile to present (at least briefly) the topic not only in the context of Benedictine orders, but also others. In addition, this section was quite poorly supported by the literature. 

2. The title includes an observation about changes in tourism trends - a very interesting issue, but also quite complicated. The author focused on some trends that are currently observed at the abbey, without explicitly pointing out the CHANGES that have occurred there - perhaps some comparative overview? Are the changes positive or negative? Which factors determined such changes, etc.? Changes in the distribution of visitors are not yet changes in trends in general.

3. The decline in visitor numbers due to the COVID-19 pandemic is a global trend, evident and observed around the world. However, is there any information that would indicate a different situation at the Abbey? Is the new approach into the tourist space of the site a real response to changes in the interpretation of the place? Have they been forced by some special situation? Maybe it's worth discussing in the conclusion?

Finally - it is worth noting the construction of the article itself. Why was the subsection "Pastoral services, teaching" placed in the chapter on "A new approach"? It seems that this is the basic activity of abbeys and is not related to the new approach and tourism in general.

In conclusion - it needs to be restructured a bit and increased emphasis on the theoretical basis explaining the research methods and goals used.

Author Response

  1. We have expanded the literature review section with 15 new literature connecting to religious tourism and economic activity of other orders of monks. (line 111-131)
  2. In the conclusion, we present in detail the changes in tourism trends regarding the abbey following the regime change. (line 645-686)
  3. We have cleared up the misunderstanding regarding pastoral activity as „new approach” (line 541-543)

Reviewer 2 Report

The text is undoubtedly an interesting case study of religious heritage in the context of cultural tourism.

In my opinion, there is a lack of context of heritage research and cultural tourism. I suggest the author use the available literature on religious heritage e. g. 

James P. Niessen, Catholic monasticism, orders, and societies in Hungary, In: Monasticism in Eastern Europe and the Former Soviet Republics, ed. Ines Angeli Murzaku, Routledge, London, 2015;

https://doi.org/10.1080/15378020.2021.2024785

and https://doi.org/10.1163/9789004363908_017

It is worth expanding (lines 143-146) and adding a paragraph indicating the context and importance of cultural tourism (religious tourism) in Hungary, e.g.

Anna Irimiás, Gábor Michalkó, Religious tourism in Hungary – an integrative framework, Hungarian Geographical Bulletin 62 (2) (2013), 175–196

Wiltshier, P. and Clarke, A. (2012) ‘Tourism to religious sites, case studies from Hungary and England: exploring paradoxical views on tourism, commodification and cost-benefi ts’, Int. J. Tourism Policy, Vol. 4, No. 2, pp.132–145

The text requires linguistic and editorial correction cf. lines: 203, 328, 373.

I suggest improving and standardizing the quality of graphs, figures and their descriptions (lines 203; 278-280; 317-319; 375,482).

In lines 385-388 it is necessary to mark in the appropriate font which passage is a quotation.

On line 385: "For 2022, there were high hopes. . . "

Author Response

  1. We have expanded the literature review section with 15 new literature connecting to religious tourism and economic activity of other orders of monks. (line 111-131)
  2. In the conclusion, we present in detail the changes in tourism trends regarding the abbey following the regime change. (line 645-686)

Reviewer 3 Report

In the Reviewer`s opinion, an outline of the phenomenon of religious tourism (respective: heritage tourism) and an indication of recent research on this phenomennon worldwide or - at least - in Hungary need to be added in the introductory section of the article.  While the historical development of the analyzed monastery is elaborated systematically and refers to rich sources, the second main context of research and publication: the tourist offer and exploration (or: creation of tourist experiences) of the religious heritage sites (as well as living religious communities)  has been completely omitted.  The paper does not refer at all to either worldwide or local (hungarian) professional literature on this branch of tourism. The collected, presented  and analyzed data on tourist visits to the abbey  and on the methods and techniques of heritage interpretation, developed and used in this place  introduce new knowledge and position the monastery as  an important  tourist attraction as well as the environment of creation and  reception of heritage experience. Although the lack of the references to contemporary religious tourism and to creation of heritage experiences deprives the author`s own research of the necessary context, suspend the conclusions in a vacuum and makes comparative analysis difficult.  

Author Response

(The authors gave the same response as above.)

Round 2

Reviewer 1 Report

The author has made appropriate additions and clarifications, most of which satisfy me. Unfortunately, I still miss the clearly declared explanation regarding the "new approach" contained in the title. It seems to me that it would be appropriate to strongly indicate what new approach have been shown in this submission - preferably in the conclusion chapter.
I suggest briefly but clearly add this to the summary.

Author Response

Thank you very much for your useful comments and suggestions. I added a new paragraph to the end of the study (line 695-703), which containes the explenation of the new approach regarding the touristic offer of the Tihany Benedictine monastery.

Reviewer 3 Report

Recommend to publish

Author Response

Thank you for work and really useful comments and suggestions!